# A Structural View at Vaccine Development against *M. tuberculosis*

**DOI:** 10.3390/cells12020317

**Published:** 2023-01-14

**Authors:** Maria Romano, Flavia Squeglia, Eliza Kramarska, Giovanni Barra, Han-Gyu Choi, Hwa-Jung Kim, Alessia Ruggiero, Rita Berisio

**Affiliations:** 1Institute of Biostructures and Bioimaging, IBB, CNR, 80131 Naples, Italy; 2Department of Pharmacy, University of Naples “Federico II”, 80131 Naples, Italy; 3Department of Microbiology, and Medical Science, College of Medicine, Chungnam National University, Daejeon 35015, Republic of Korea

**Keywords:** tuberculosis, vaccine, protein, structure, immune response

## Abstract

Tuberculosis (TB) is still the leading global cause of death from an infectious bacterial agent. Limiting tuberculosis epidemic spread is therefore an urgent global public health priority. As stated by the WHO, to stop the spread of the disease we need a new vaccine, with better coverage than the current *Mycobacterium bovis* BCG vaccine. This vaccine was first used in 1921 and, since then, there are still no new licensed tuberculosis vaccines. However, there is extremely active research in the field, with a steep acceleration in the past decades, due to the advance of technologies and more rational vaccine design strategies. This review aims to gather latest updates in vaccine development in the various clinical phases and to underline the contribution of Structural Vaccinology (SV) to the development of safer and effective antigens. In particular, SV and the development of vaccine adjuvants is making the use of subunit vaccines, which are the safest albeit the less antigenic ones, an achievable goal. Indeed, subunit vaccines overcome safety concerns but need to be rationally re-engineered to enhance their immunostimulating effects. The larger availability of antigen structural information as well as a better understanding of the complex host immune response to TB infection is a strong premise for a further acceleration of TB vaccine development.

## 1. Introduction

*Mycobacterium tuberculosis* (*Mtb*), the etiological agent of Tuberculosis, is still one of the most life-threatening pathogens. In 2021, *Mtb* infected 10.6 million people and killed 1.6 million, with cases of multidrug-resistant TB having risen to 450,000 globally [1]. Tuberculosis (TB) is airborne-transmitted and mainly affects the lungs, although it can also affect other districts. Following the exposure to *Mtb*, 5–10% of infected individuals develop active TB disease within the first five years. In the remaining 90%, the infection remains latent retaining a persistent risk of developing active TB disease throughout their lifetime [2]. Furthermore, co-infections with other pathogens, such as HIV, promote the progression to the active disease, with a 20-fold increase of the risk of latent TB reactivation [3].

The success of *Mtb* fully relies on its ability to survive and persist within the hostile environment of macrophages, establishing a dynamic equilibrium with host immune system [4]. This pathogen’s remarkable resilience and infectivity is largely due to its unique cell envelope, comprising of four main layers: (i) the plasma membrane or inner membrane (IM), (ii) the peptidoglycan–arabinogalactan complex (AGP), (iii) an asymmetrical outer membrane (OM) or ‘mycomembrane’, that is covalently linked to AGP via the mycolic acids, and (iv) the outermost capsule [5,6]. The outer membrane has an unusual, waxy coating on its cell surface primarily due to the presence of mycolic acid that are oriented perpendicular to the plane of the membrane and provide a truly special lipid barrier responsible for many of the physiological and disease-inducing aspects of the bacterium [5,6]. Modifications of this complex cell envelope have been associated to the remarkable ability of *Mtb* to resuscitate from its state of dormancy even after 10 years [7,8].

Despite intensive research efforts, *Mtb* pathogenicity mechanism is still poorly understood, and thus diagnosis and therapy for TB remain a challenge. The standard TB treatment consists of a combination of isoniazid, rifampicin, ethambutol, and pyrazinamide, followed by a combination of isoniazid and rifampicin only [9]. Nevertheless, inadequate treatment, development of drug resistance and delays in diagnosis contribute to the severity and mortality of the disease [10,11]. Therefore, preventing TB disease rather than curing the infection has been the primary target for vaccine development.

Historically, vaccines were developed empirically through an ‘isolate, living or not-living, and inject’ approach [12]. In the case of TB, the Bacillus Calmette-Guérin (BCG) vaccine, which was developed in 1921 with this approach, offers only a partial and variable protection in infants and has a little impact in controlling the pulmonary infection in all other age groups, as adolescents and adults [13]. In the last three decades, several strategies and a broad range of vaccine technologies have been tested to improve or replace BCG- vaccine [14]. The development of a vaccine may start with the identification of the antigenic components of the microorganism capable of stimulating a protective immune response. However, these approaches are time-consuming and labour-intensive. Several barriers, like a highly complex pathogen and host immune response have compromised an efficacious vaccine development [15]. Substantial improvements in bioinformatics and in genome-based approaches have enormously contributed to the *in-silico* screening and the identification of promising vaccine antigens without the need to grow the microorganism and purify each antigen. This genome-based approach is often described as reverse vaccinology (RV)[16] and has been applied for the first time in early 1990′s for the development of a vaccine against the serogroup B strains of *Neisseria meningitidis* (MenB) [17,18]. RV can help to discover novel antigens that are likely to be missed by conventional approaches and has contributed to produce new knowledge and antigen candidates to be used in a novel vaccine [19,20]. More recent advances in bioinformatic tools and structural biology has allowed for even more rational strategies for vaccine development, using structural biology combined to computational approaches (i) to predict B- and T-cell epitopes, (ii) to evaluate their conservation among other pathogenic antigens, and (iii) to predict immunogenicity, toxicity and solubility [21].

A World Health Organisation (WHO) initiative, The End Tuberculosis Strategy, set ambitious targets for 2020–2035, 90% reduction in TB incidence and 95% reduction in TB deaths by 2035, compared with 2015 [22] and, beyond, the elimination of the disease as a global health problem by 2050. However, without effective TB vaccines, we are unable to suppress the global TB emergency. Currently, BCG is still the only vaccine approved to prevent TB, despite its known inefficacy. However, several vaccines against TB are in the pipeline, including live attenuated whole-cell vaccines, inactivated whole-cell vaccine, adjuvanted protein subunit vaccine, and viral-vectored vaccines [23]. Among these, subunit vaccines are extremely promising candidates, as they overcome safety concerns and optimise antigen targeting. However, a comprehensive knowledge of TB-induced innate and adaptive immune responses is key to the design of effective antigens able to stimulate more types of immune mechanisms. Also, strong efforts must be dedicated to antigen re-engineering to enhance their immune-stimulating effects. In this review, we report the current knowledge of TB induced immune responses and provide a panoramic of the current stage of TB vaccine development, by describing vaccines which are currently in clinical trials. Also, we specifically focus on our previous contributions in the development of subunit vaccine antigens against TB and the role and potential of structural vaccinology in further improving subunit vaccines to finally reach the challenging promise of WHO to quickly eradicate TB.

## 2. Immune Response to TB Infection

*Mtb* spreads from one person to another through tiny droplets released into the air with coughs and sneezes. The bacilli reach the pulmonary alveoli through the bronchial tree and start to infect the alveolar macrophages. Typically, within a few weeks the immune system is able to stem the infection, confining the infected macrophages to a kind of bulwark made up of aggregates of immune cells, called granuloma [24]. This peculiar formation in the lung, with infected macrophages in the center and lymphocytes, stem cells, and epithelial cells in the marginal zone, is a typical sign of primary response to *Mtb* infection. The bacilli inside the estimated 5 to 10% of people exposed to *Mtb* develop an active form (ATB) [2]. The natural granuloma can be killed or survive in a quiescent state for several months. While the majority of individuals exposed to *Mtb* are able to control infection in the form of Latent TB (LTBI), an arsenal against *Mtb* and other pathogens is the host immune system, which acts through the concerted actions of the innate and adaptive immunity.

### 2.1. Innate Immunity against Mtb

The major players of innate immunity against *Mtb* include airway epithelial cells (AECs), macrophages, neutrophils, natural killer cells (NKs), dendritic cells (DCs), mast cells and the complement [25].

AECs represent a physical barrier located in the bronchial tree and are the first cells to meet *Mtb*. AECs can sense the bacilli through pattern recognition receptors (PRRs), whose activation leads to the production of inflammatory cytokines and activation of mucosa-associated T cells, which in turn stimulate the production of inferferon-γ (IFN-γ) and tumor necrosis factor (TNF)-α [26]. *Mtb* is phagocytosed by alveolar macrophages, which can eradicate mycobacteria through several mechanisms, including autophagy, production of oxygen and nitrogen components and cytokines, and acidification of the phagosome [27]. Recognition between bacilli and macrophages occurs through the interaction between pathogen-associated molecular molecules (PAMPs) of *Mtb* (such as, lipoproteins, glycolipids and carbohydrates) and macrophage PRRs (such as TLRs and NLR) [26]. This interaction induces the expression in macrophages of several inflammatory cytokines, including IFN-γ, IL-12, IL-1β, and macrophage inflammatory protein-1α (MIP-1α/CCL3). These cytokines recruit blood polymorphonuclear leukocytes (PMN), additional mononuclear leukocytes and T lymphocytes to the site of infection [28]. The result is the accumulation of immune cells around the foci of infected cells to form the granuloma, whose role is to control and limit bacillary dissemination. Indeed, its persistence depends on the local production of IFN-γ by antigen-specific T lymphocytes, which respond to the presence of *Mtb* antigens and lead to conspicuous activation of macrophages. If optimal, this balance leads to complete bacillary control or latency [28].

Structurally, granulomas are compact and organised aggregates, with macrophages surrounding the site of infection and other immune cells, including neutrophils, DCs, NK cells, located at their periphery (Figure 1A). Neutrophils are recruited in situ by pro-inflammatory chemokines and cytokines and alarmins (S100A8/A9 proteins) and play a complex role in the immunopathology due to *Mtb* infection [29]. Indeed, factors released by neutrophils during respiratory bursts, such as elastase, collagenase, and myeloperoxidase, indiscriminately damage both bacilli and host cells. Furthermore, other proteins (arginase and matrix metalloprotein 9), released by neutrophils lead to damage of lung parenchyma [25]. Also, lymphocyte apoptosis can be induced by the programmed death ligand-1 (PD-L1) expressed on the neutrophil cell membrane [26,30]. NK cells are large granular lymphocytes which recognise the target nonspecifically and without the aid of major histocompatibility complex (MHC) molecules. They cause lysis or apoptosis of target cells and, when stimulated with cytokines of the interferon class (such as IFN types α, β, and γ, IL-2 or IL-12), can increase their cytotoxic power. Activated NK cells are able to bind different components of the bacterial cell wall of *Mtb*, through their NKp44 receptor [27]. In addition, NK cells can recognise stress molecules overexpressed on the surface of host cells after *Mtb* infection. Following the recognition event, NK cells can act on *Mtb* through canonical cytotoxic mechanisms, such as the production of cytoplasmic granules containing perforin, granulysin and granzyme, or indirectly, stimulating and activating macrophages.

DCs are the most interesting cellular elements of the immune system because their function straddles the line between innate and adaptive immunities. Indeed, since T lymphocytes are not able to recognise directly complex protein antigens, DCs are required to process the protein antigens into smaller fragments and conjugate these antigenic peptides to MHC I or MHC II complexes. During *Mtb* infection, DCs bind and internalise the bacilli through the receptor DC-SIGN, (Dendritic Cell-Specific Intercellular adhesion molecule-3-Grabbing Non-integrin) also known as CD209 (Cluster of Differentiation 209) [26]. This is a C-type lectin receptor present on the surface of the DCs [31]. Once the MHC has bound the protein antigens, it is recognised by CD8 and CD4 T cells and activate an adaptive immune response. Dendritic cells also have a great capacity for cytokine production and stimulate T-cell differentiation. For example, secreting IL-12 and IL-6, DCs differentiate the CD4 T cell into a T helper (Th) 1 cell and Th 2, respectively [31].

### 2.2. Adaptive Immunity against Mtb

As mentioned above, *Mtb*-infected APCs (macrophages and DCs) secrete cytokines that include IL-12, IL-23, IL-7, IL-15 and TNF-α, and present antigens to several T-cell populations, including CD4+ T cells (MHC class II pathway) and CD8+ T cells (MHC class I pathway) (Figure 1B) [32]. Activated APCs stimulate the T cell receptor (TCR) resulting in T cell proliferation. Following TCR activation by APCs, CD4+ T cells differentiate into either T helper cells (Th), which secrete diverse effector molecules contributing to cell-mediated or humoral immune responses, inflammation or immune-regulation (Figure 1). The importance of CD4+ and CD8+ T cells is principally related to their IFN-γ production, which is particularly relevant in the early stages of *Mtb* infection [33]. Indeed, IFN-γ activates macrophages in conjunction with TNF-α to effect killing of intracellular mycobacteria through reactive oxygen and nitrogen intermediates. In addition, CD8+ cytotoxic T cells can kill intracellular mycobacteria through granulysin and perforin-mediated pathways (Figure 1B).

The adaptive response mediated by T lymphocytes plays a key role in eliminating *Mtb* whereas a minor role is attributed to humoral or antibody-based immunity in preventing the infection. Humoral immunity, one of the two arms of the adaptive immune response, typically results in the generation of antigen-specific antibodies that target invading microbial pathogens or vaccine antigens. In the case of TB, there is emerging evidence that B cells and humoral immunity can modulate the immune response [34]. Indeed, B cells and antibodies are extremely important in modulating the immune response to mycobacteria through several mechanisms. For instance, B cells process and present antigens to T cells, secrete antibodies, and modulate inflammation through the production of IL-10. Also, they can stimulate Th1 responses through the production of IL-12, IFN-γ and TNF-α [35]. Different than humoral immunity, cell-mediated immunity involved in protective immune response to *Mtb* is well understood, due to the availability of animal models. Indeed, studies in mice have shown that CD4+ Th cells, CD8+ T cytotoxic cells are the most protective T cells against TB [36]. Indeed, the process of phagosome maturation is facilitated and increased by IFN-γ. In addition, within the lymph nodes, T lymphocytes undergo a process of activation and production of populations which are specific for *Mtb* antigens. Well-known immuno-dominant antigens recognised by T cells include Early Secretory Antigenic Target-6 (ESAT-6), Culture Filtrate Protein antigen 10-kDa (CFP-10), antigens 85A and 85B (Ag85A, Ag85B) and TB10.4. However, the identification and study of new antigens recognised by these T cells is critical for a better understanding of the immuno-biological characteristics of TB and for the development of new vaccines.

## 3. Vaccine Approaches against TB

Currently, the global TB vaccine portfolio includes over 16 distinct vaccine candidates at different stages of clinical development (Figure 2). These candidates fall into several categories: (i) live attenuated mycobacterial vaccine, which target either BCG- replacement at birth and/or prevention of TB in adolescents and adults, (ii) whole killed and fragmented-cell vaccines for therapeutic purposes, (iii) adjuvanted protein subunit vaccines based on one or more recombinant *Mtb* fusion proteins for preventive pre- and post-exposure, and (iv) viral-vectored vaccines, whose goal is to boost BCG-immunity.

### 3.1. Live-Attenuated Whole-Cell Vaccines

Live-attenuated whole-cell vaccines, which have been developed as prophylactic pre-exposure vaccines (priming vaccines) aiming to replace BCG vaccination in neonates, are now also being assessed as post-exposure vaccines in adults to prevent TB recurrence [37]. Major drawbacks of this technology rely on its labor-intensive and stringent quality control, resulting in increased manufacturing costs and slow response in the event of a pandemic [38]. The advantage of using live-attenuated vaccines, compared others, is their ability to mount a complex and diverse immune response due to a wide range of antigens. Indeed, this class possess a broader antigenic panel and induces an immune response that more closely resembles a natural *Mtb* infection, leading to increased long-term protection [37].

Two examples of this class, VPM1002 and MTBVAC, have advanced in clinical trials (Figure 1A). VPM1002 is a recombinant BCG (rBCG) vaccine where the urease C gene has been replaced by the listeriolysin O gene (LLO) from Listeria monocytogenes. LLO expression in VPM1002 results in the release the mycobacterial antigens into the cytosol, triggering autophagy, inflammasome activation and apoptosis [39]. VPM1002 has demonstrated increased immunogenicity, efficacy and safety in preclinical studies. Furthermore, it has successfully passed Phase I and II clinical trials in adults and infants, and it is currently under Phase III [39,40] (NCT0435685). Results to date have proven VPM1002 to be significantly safer than the BCG vaccine, with particular regards to HIV- and non-HIV exposed newborn infants in South Africa [41].

MTBVAC is a live rationally attenuated *Mtb* derivative of Mt103 strain, designed to contain two independent stable genetic deletions in the genes phoP (Rv0757) and fadD26 (Rv2930). These genes encodes of two major virulence factors in *Mtb*; PhoP is required for the transcription of several virulence genes, whereas fadD26 is essential for the biosynthesis and export of virulence-associated cell-wall lipids [42]. To date, MTBVAC is the first and only candidate of its nature enrolled in human clinical trials; it has completed Phase I and Phase II in adults and new-borns, proving to be safe and more effective compared to BCG [43], whereas is ongoing Phase III efficacy trial in infants in TB-endemic countries (NCT04975178).

### 3.2. Inactivated Whole- and Fragmented-Cell Vaccines

Inactivated vaccines utilise inactivated whole mycobacteria or cleavage fragment thereof, to elicit an immune response against multiple *Mtb* antigens. These vaccines have been developed for both pre- and post-exposure strategies and amongst them, RUTI, DAR-901 and MIP are currently in clinical trials.

RUTI contains detoxified fragments from *Mtb* cell wall delivered in a liposome suspension [44]. It is currently proposed as a therapeutic vaccine to be used in conjunction with the standard antibiotic treatment. Combined Phase I/II clinical trial demonstrated reasonable tolerability of this vaccine in persons with LTBI [45,46], whereas small Phase II trials are ongoing to evaluate the efficacy, clinical safety and immunogenicity of RUTI as adjunctive therapy for drug sensitive and rifampicin-resistant tuberculosis (NCT04919239, NCT02711735, NCT05136833).

MIP and DAR-901 exploit a different strategy, using inactivated organisms belonging to non-pathogenic and environmental mycobacterial species. The use of whole-cell inactivated vaccines is based on the assumption that antigens shared between *Mtb* and these saprophytic mycobacteria are relevant to protective immunity, albeit lacking the harmful components of *Mtb* [43]. DAR-901, a broth-grown preparation derived from *M. obuense*, has recently completed the phase IIb trial to assess its capability as a booster vaccine to prevent TB in BCG-primed adolescents in Tanzania [47]. Finally, MIP contains heat-killed *M. indicus pranii*, a non-pathogenic mycobacterium closely related to *M. avium* [48]. Potential application of MIP as a booster to BCG vaccine for efficient protection against TB has been demonstrated in a Phase III trial [49].

### 3.3. Viral-Vectored Vaccines

Viral vector vaccines use viruses (viral vectors) to deliver the genetic sequence encoding for the antigen. Several viral vectors have been tested for vaccine delivery. Limitations of viral-vectored vaccine are related to their immunotoxicity, that affect vaccine safety and efficacy. Indeed, an accurate vector design is mandatory to suppress inflammatory responses and for a mediate gene therapy [50]. Nevertheless, the use of viral vectors for the production of prophylactic vaccines is rapidly increasing due to the versatility of the manufacturing process and the possibility of a rapid distribution during emergency event [51].

Viral-vectored TB vaccines are hitherto in Phase I clinical trials, and other promising candidate vaccines at an earlier stage of development are being evaluated in preclinical animal models (Figure 2). Among them, MVA vectors have been extensively tested as prophylactic vaccine platform in endemic settings. MVA (Modified Vaccinia Ankara) is a recombinant replication-deficient modified vaccinia virus, that allows insertion of large immunogenic sequences, up to 10 kbp, and is efficient at induction of specific T cell responses. MVA expressing the *Mtb* Ag85A antigen (MVA85A), was the first prophylactic TB vaccine candidate to be evaluated in a Phase 2b efficacy trial. However, the study showed the vaccine was safe but unable to improve the efficacy of BCG [52]. Consequently, another vector, ChAdOx1 (Chimpanzee Adenovirus-vectored Oxford 1) has been used in combination with MVA85A. ChAdOx1 is a simian adenoviral vector, which was modified to avoid its replication. The ChAdOx1 vectored vaccine ChAdOx1.85A, encoding Ag85A, has completed Phase I and IIa clinical studies as a booster to BCG vaccination in combination with MVA85A (NCT01829490, NCT03681860). Preclinical murine studies have demonstrated ChAdOx1.85A to be consistently protective when used as part of a BCG-ChAdOx1 85A-MVA85A immunisation regime [52]. Furthermore, Phase I clinical trial showed that ChAdOx1.85A prime—MVA85A boost is well tolerated and immunogenic in healthy UK adults [53]. Also, ChAdOx1.PPE15, encoding for both Ag85A and PPE15 antigens, is in preclinical evaluation and has been shown to confer superior protection to BCG alone [54].

Ad5 Ag85A is a recombinant replication-deficient human adenovirus vector (Ad5, adenovirus type 5) expressing the Ag85A antigen. Preclinical studies showed that Ad5 Ag85A gives a markedly improved protection over BCG and is suitable for immunization via the respiratory route. Ad5 Ag85A proved to be safe also in humans, through intramuscular vaccination, and stimulated polyfunctional T cell responses more potently than BCG [55].

Other viral-vectored vaccines in clinical trials utilise the influenza A vector platform, through mucosal administration. TB/FLU-01L is based on an attenuated replication-deficient influenza virus expressing the mycobacterial antigen ESAT-6, whereas for TB/FLU-04L the viral vector was designed to express Ag85A and ESAT-6. Phase I trials showed that these candidate vaccines, administered intranasally or sublingually, are safe and immunogenic in BCG-vaccinated healthy adults (NCT03017378, NCT02501421).

### 3.4. Protein Subunit Vaccines

Subunit vaccines candidates contain purified parts of the pathogen (either a protein or a polysaccharide) that elicit immunogenic host response and are produced using recombinant DNA technologies [56]. Although they are specific and safe, the limited number of antigens present in subunit vaccines reduce their capacity to stimulate a broad immune response compared to the previously described vaccines, composed of live attenuated or killed microorganisms. Therefore, these vaccines need to be engineered to enhance their immunogenicity, decrease the needed antigen dose, ensure a targeted delivery, and optimise antigen delivery and interaction with the immune cells [57]. Also, the use of adjuvants, whose mechanisms are described in next paragraph, is necessary to enhance their induced immune response. In general, these subunit vaccines are given as boosters after a BCG prime, to improve BCG-mediated protection or to increase the duration of the protection offered by BCG prime vaccine; but they are also being evaluated for their therapeutic value.

Subunit vaccines against TB currently in clinical trials include five candidates (Figure 2) [58]. Among these, H56:IC31 is an adjuvanted vaccine that contains three *Mtb* antigens, Ag85B, ESAT-6 and Rv2660c, with the adjuvant IC31 as a TLR-9 agonist (Figure 2B). Phase I studies have shown that vaccination with H56:IC31 is safe and immunogenic in BCG-vaccinated adults with or without *Mtb* infection [59]. Moreover, other investigations have proved its therapeutic potential for the treatment of both drug-sensitive and multi-resistant TB [60]. ID93 contains four different *Mtb* antigens, which include two predicted outer membrane proteins, Rv1813 and Rv2608, and two secreted proteins, Rv3619 and Rv3620, belonging to the ESAT6 family. Over the last half decade, many studies have been conducted on ID93+GLA-SE, as those to assess its immunogenicity in BCG-vaccinated healthy adults [61]. This vaccine has been evaluated in Phase IIa for the prevention of recurrence in HIV-uninfected patients who were diagnosed with drug-sensitive pulmonary TB [62]. M72/AS01_E_ is composed of a recombinant fusion protein of two *Mtb* antigens, Mtb32A and Mtb39A, in combination with the AS01_E_ adjuvant system. Phase II studies have shown that this vaccine provide 54.0% protection against active pulmonary disease without any safety issues involved. Also, vaccination with M72/AS01_E_ elicits both antibody and T cell-mediated immune response and has shown to be efficacious for up to 3 years [63].

AEC/BC02 is composed of the recombinant *Mtb* antigen Ag85B, the fusion protein ESAT6-CFP10 (EC), and a complex adjuvant system BC02. This candidate has completed Phase I clinical trials, whereas Phase IIa is ongoing (NCT05284812). This vaccine also demonstrated a good protective effect in a guinea pig latent infection model, reducing the degree of the viable bacterial load in the spleen and lungs [64]. AEC/BC02 has also been proven to have a therapeutic effect after chemotherapy with isoniazid and rifapentine [65]. Finally, GamTBvac is a new BCG booster candidate vaccine containing Ag85A and ESAT6-CFP10 *Mtb* antigens, together with the CpG ODN adjuvant, formulated with dextrans. This is the most advanced TB subunit vaccine, currently in Phase III trials [66].

### 3.5. The Contribution of Adjuvants

Adjuvants are involved in the enhancement and/or shaping antigen-specific immunity, mostly through the depot formation in the injection site, protection of the antigen from degradation and the stimulation of immune system [67,68]. Unfortunately, traditional development of vaccine adjuvants has been recognised as one of the slowest processes in medicine history [69], making alum the only licensed adjuvants for decades [70]. Most implemented adjuvants in TB vaccine development include (i) emulsions and liposomes, like GLA and AS01, (ii) fusion proteins, like IC31 and (iii) Cytosine phosphate guanosine (CpG) nucleotides, polysaccharides [71] (Figure 2A).

Glucopyranosyl Lipid Adjuvant (GLA) is a synthetic derivative of lipopolysaccharide (LPS) which works as an agonist of TLR4. Although GLA possesses immunostimulatory activity like LPS, its toxicity is highly reduced. GLA exists in a formulation with oil-in-water squalene emulsion named GLA-SE [72]. As reported above, GLA-SE with protein ID93 is in Phase IIa of clinical trials, where its tolerability and elicitation of the T-cell response were shown to be promising [62,73]. Similarly, AS01 is a liposome-based adjuvant with two immunostimulants, MPL and QS-21, and cholesterol to decrease haemolytic action of QS-21. In the formulation MPL acts as TLR4 activator and stimulator of the signalling pathway that leads to activation APCs, whereas QS-21 is a natural saponin with two fractions of isomeric triterpene glycosides form tree Quillaja Saponaria. Although direct action of QS-21 is not clear it was suggested that it acts as activator of both APCs and Th1 immune response [74]. AS01 in the subunit vaccine M72:AS01E is currently in Phase IIb clinical trial, with promising results and efficacy of app. 50% in patients with latent TB were it elicits protection against progression into pulmonary TB for at least three years [63,75].

IC31 is a two-component adjuvant in clinical evaluation, constituted by positively-charged peptide KLK (KLKL5KLK) and a phosphodiester-backboned DNA oligonucleotide consisting of repeats of the dinucleotides deoxyinosine and deoxycytosine (ODN1a) in molecular ratio 25:1 respectively [76]. Here KLK works as a carrier to translocate ODN1a into the cell without cell membrane permeabilisation. Inside the cell, ODN1a acts as an agonist on the TLR9 receptor, that is normally activated by bacterial DNA [77]. This adjuvant in formulation with H56 is under Phase I clinical evaluation. Results of previous studies confirmed safety and tolerability of H56:IC31 in humans, with an antigen-specific CD4+ T cell response [78,79].

Cytosine phosphate guanosine (CpG) oligodeoxynucleotides (ODN) are short, single-stranded synthetic DNA molecules, whose sequence is often observed in bacterial and viral DNA. CpG are Toll-like receptor agonists, specifically recognized by Toll-like receptors 9 [4,5]. CpG ODN is currently used as one of the adjuvant components of GamTBvac, the sole subunit vaccine hitherto in Phase III trials. CpG ODN are also contained in the BC02 (BCG CpG DNA compound adjuvants system 02) adjuvant, composed of BCG CpG ODN with Al(OH)_3_ salt, used for delivery. Its safety and strong adjuvant efficacy have been effectively verified in clinical trials and it belongs to the AEC/BC02 vaccine formulation, currently in Phase IIa trials [80].

An interesting approach for adjuvant development is also the use of polysaccharides as starch or cellulose. In fact, raw starch was proven to be safe, biocompatible, and biodegradable as an adjuvant. Starch is an α-glucan with molecular size of the molecules that allows it to mimetic glycogen-like α -glucan, important mycobacterial cell wall component. Administrated starch is recognised by C-type lectin DC specific ICAM-3-grabbing nonintegrin, specific for α-glucan and can elicit specific immune response [81,82]. Starch was also considered as delivery system and a step forward toward mucosal TB vaccine [83].

## 4. Structural Vaccinology as a Tool to Enhance Antigenicity of TB Subunit Vaccines

Structural vaccinology (SV) or structure-based antigen design is a rational approach that uses three-dimensional structural information to design novel and enhanced vaccine antigens. As previously mentioned, this strategy is crucial to the development of effective subunit antigens.

Over the last decade, SV was shown to be successful in improving vaccine candidates’ immunogenicity through protein structural modification. The first proofs-of-concept were achieved for the meningococcal fHbp antigen and the group B streptococcal pilus antigen, both through the engineering of multiple immunodominant epitopes in one single molecule able to induce broad immune responses against different protein variants [84,85]. Structure-based antigen design has also been used in vaccine development against viral pathogens, including HIV, RSV, MERS-CoV, SARS-CoV2, Influenza A and B [86,87,88,89,90].

The typical SV approach, reported in Figure 3, involves the determination of the three-dimensional structure of the antigen or antigen–antibody complex using structural biology tools. This is achieved by combining the benefits of RV, with advancements in X-ray crystallography, NMR spectroscopy and single particle Cryo-Electron Microscopy, and novel computational approaches, including Artificial Intelligence (AI). Experimental data on protein structures and AI predictions can be exploited and used to improve the reliability of predictions of immunogenicity, toxicity, allergenicity, using a variety of available software.

A vast number of bioinformatic tools have become available to assist the scientists in identifying the immunogenic properties of potential antigens and reducing the time of research [91] (Table 1). While designing an antigen, a comprehensive sequence analysis is needed to evaluate the level of antigen conservation among serotypes and, in luckiest cases, among different pathogenic bacteria. During this process, an effort has to be made to skip those antigens which are also conserved in non-pathogenic strains, especially those belonging to the microbiota. Indeed, it has been recently shown that the composition and function of the gut microbiota are crucial in modulating immune responses to vaccination [92], especially in early life immunity [93].

Indeed, immunological and bioinformatic knowledge are synergistically combined with the structural insights allowing the re-design of vaccine antigens through protein engineering strategies. Finally, re-engineered antigens or epitopes are tested for their safety and efficacy in cell-based assays and in vivo. Based on the preclinical findings, the candidate vaccines can be redesigned to improve their immunogenicity and efficacy.

Structural vaccinology represents a powerful tool for the rational design or modification of vaccine antigens allowing (i) to selectively enhance antigen specific immunogenic properties, (ii) to combine more immune stimulation strategies by conjugating more antigens and (iii) to develop antigens with multi-epitope presentation [94,95,96]. SV can be an answer for today’s challenges, aiming also at the optimisation of vaccines in the terms of HLA-differences [97,98] or responsiveness in various age groups, especially mature individuals [99].

**Table 1 cells-12-00317-t001:** A collection of bioinformatic tools for antigen prediction.

Tool	Function	Method	Reference
VaxiJen v2.0	Prediction of protective antigens and subunit vaccines	Sequence-based	[100]
BepiPred 3.0	Prediction of potential B-cell epitopes	Sequence-based	[101]
DiscoTope 2.0 IEDB	Prediction of potential B-cell epitopes	Structure-based	[102]
ElliPro IEDB	B cell epitope prediction, based on solvent-accessibility and flexibility	Structure-based	[103]
SVMTriP	Prediction of protein regions preferentially recognised by antibodies	Sequence-based	[104]
Peptide binding to MHC class I molecules IEDB	MHC Class I epitope prediction	Sequence-based	[105]
Peptide binding to MHC class II molecules IEDB	MHC Class II epitope prediction	Sequence-based	[106]
NetMHCpan4.1 NetMHCIIpan4.0	Prediction of CD8 and CD4 T cell epitopes	Sequence-based	[107]
AllergenFP v.1.0 AllerTop v.2.0	Allergenicity prediction	Sequence-based	[108,109]
ToxinPred	Toxicity prediction	Sequence-based	[110]
Clustal Omega	Multiple sequence alignment using seeded guide trees and HMM profile-profile analysis	Sequence-based	[111]

The SV approach has also shown to be a useful tool to design multi-epitope vaccines against TB. Structure-based epitope mapping of the *Mtb* secterory antigen MTC28 allowed the identification of its immunogenic part, by coupling X-ray crystallography with experimental and computational epitope mapping [112]. Computational tools were also used for the design of a multi-epitope using ESAT-6 and Ag85B as two main secretory antigens in the active Phase of *Mtb* infection, and Fcγ2a as targeted delivery system [113]. Furthermore, an integrated computational approach was used to design a multi-epitope vaccine based on two antigenic proteins, diacylglycerol acyltransferase and ESAT-6-like protein [114]. Finally, the use of nanoparticles allows for improved antigen stability and immunogenicity, due to multiepitope presentation. Also, growing evidence is accumulating on the advantages of nanoparticles in targeted delivery and slow release [115].

## 5. Novel Promising Subunit Vaccines against TB

Apart from the subunit vaccines in clinical trials, more subunit antigens are being proposed in preclinical studies, like H107 and CysVac2. Both vaccines are fusion proteins and are strongly immunogenic when are adjuvanted [116,117]. Also, several proteins have been shown to possess immunomodulatory properties and constitute a reservoir of optimum candidates for TB vaccine development. In this framework, we have investigated a panel of vaccine antigens exploiting diverse typologies of immune stimulations (Table 2). These proteins play different roles in *Mtb* biology and different cell localisations. Indeed, some of them are periplasmic and involved in its complex cell wall processing, through the degradation of its peptidoglycan. Others play important role in cytosolic functions, such as ribosome recycling factors, Universal Stress Proteins (USP) etc. However, independent of their roles and cell localisation, they can stimulate *Mtb* innate and, in some cases, adaptive response. We believe that a successful vaccine strategy must, beside using structural information for antigen improvement, combine different immune stimulation mechanisms. Therefore, examples of conjugated antigens exploiting different immune stimulation mechanisms are given in this section. Importantly, the recent availability of AI structures, due to the release of the AlphaFold software, makes more SV studies possible also when experimental structural data are not available and is a potential hope for the future [118].

### 5.1. Peptidoglycan Processing Enzymes as Dendritic Cell Stimulators

RpfB and RpfE belong to the family of five secreted proteins, denoted as Resuscitation Promoting Factors (Rpfs A-E). These proteins play a role in *Mtb* resuscitation through the hydrolysis of peptidoglycan (PGN), the major constituent of *Mtb* cell wall [121,122,123,132,133]. Structurally, all Rpfs contain a homologous catalytic domain (Figure 4), which shares structural similarity with lysozyme [121,134]. RpfB is the largest and most complex of the five Rpf proteins encoded by *Mtb* and the sole to produce delayed reactivation from chronic TB [135]. This enzyme acts by hydrolysing the glucosidic part of PGN, which is formed by glycan chains of β-(1-4)-linked N-acetyl glucosamine (NAG) and N-acetyl muramic acid, cross-linked by short peptide stems [136]. In *Mtb*, these peptide stems are called DAP-type, as they contain meso-diaminopimelic acid (DAP) residues (Figure 5). Whereas hydrolysis of NAM-NAG chains of PGN is operated by Rpfs, hydrolysis peptide crosslinks is due to another cell wall hydrolase, the NlpC/P60 endopeptidase as RipA (Resuscitation promoting factor interacting protein A). This enzyme governs cell division and likely resuscitation and has a remarkable effect on the bacterial phenotype, as ripA depletion strains in *M. smegmatis* exhibit an abnormal phenotype, consisting of branching and chaining bacteria [7,137,138]. The name of RipA is due to its feature of co-localising with RpfB at bacterial septa [139], with their combined actions being synergistic in PGN hydrolysis [140,141]. Consistently, it was shown that a product of RpfB and RipA joint enzymatic action promotes the resuscitation of dormant mycobacteria [133].

Beside this role in PGN remodelling, RpfB was shown to induce activation of DCs mediated by direct binding of RpfB to TLR4, followed by MyD88/TRIF signalling, with an important role of the G5 domain (Figure 6A) of RpfB in TLR4 binding [124]. Importantly, RpfB-treated DCs activated naïve T cells and induced T cell proliferation, indicating that this protein contributes to Th1 polarisation of T cell immunity. Notably, unlike with LPS, RpfB acts as a specific recall antigen. Indeed, the novelty of RpfB lies in its ability to incite innate and adaptive immunity simultaneously with a self-adjuvanting property-activating DC toward Th1 polarised-naïve/memory T cell expansion in a TLR4-dependent cascade [124]. With this mechanism, RpfB, and in particular its G5 domain, succeeds in inducing long-lasting Th1 memory, which is the central feature of a successful TB vaccine [124]. We previously characterised the crystal structure of a major portion of RpfB, including its catalytic, D5 and first DUF348 domain (pdb code 5e27), and modelled the entire structure based on the sequence homology among the three DUF348 domains [122,123]. The recent advance of Artificial Intelligence confirmed the modelled structure (Figure 6A). As shown in Figure 6A, RpfB adopts a flat S shape structure, which we previously proposed as key to its adhesion to the peptidoglycan layer [122,123]. Indeed, with this structure, the catalytic domain of RpfB is properly oriented towards PGN NAM-NAG chains whereas the G5 and DUF348 domains contribute to PGN adhesion [123].

Another Rpf interesting immunomodulating function is RpfE. Indeed, we showed that RpfE-matured DCs stimulate naïve T cells and antigen specific T cells to produce IFN-γ and IL-17 [120]. In this process, prostaglandin E2 (PGE2), which is a key biological mediator in the defense against *Mtb* infection, plays an important role in Th1 and Th17 cell responses due to RpfE-activated DCs. Indeed, mice administered intranasally with PGE2 displayed RpfE-induced antigen-specific Th1 and Th17 responses with a significant reduction in bacterial load in the lungs. These results clarified the antigenic mechanism of RpfE DC-mediated immunogenicity, as they showed that RpfE-matured DCs produce PGE2, which in turn induces Th1 and Th17 cell differentiation with potent anti-mycobacterial activity [119]. Different than RpfB, RpfE possesses a rather simple structure, formed by its catalytic lysozyme-like domain preceded by a Pro/Ala rich region (Figure 4). Of this protein, only the crystal structure of its catalytic domain (residues 98–172, pdb code 4cge) is hitherto known [142]. Indeed, its N-terminal region contains, beside a signal peptide (residues 1–29), a region rich in proline and alanine (residues 30–97) that is predicted, using DISOPRED3 [143], to be characterised by a high degree of flexibility. Consistently, also using the recent advance of AI, through the software AlphaFold [118], we could obtain no reliable structure, due to low pLDDT, between 50 and 70 (Figure 6B). Also, it is so far unknown which region of RpfE is mostly involved in DC maturation.

Both in the case of RpfB and RpfE, sharing DC-maturation properties, there are still large margins of improvement using SV, upon the identification of the exact immune-stimulating region and the design of more tailored recombinant antigens. Indeed, given the importance of DCs for priming T cells, new antigens can be designed to enhance DC response and to combine DC-stimulation with other strategies to boost the immune response.

### 5.2. Rv2882c and Rv2005c: Combining Dendritic and Macrophage Stimulation

Recombinant Rv2882c protein activates macrophages to secrete pro-inflammatory cytokines and express co-stimulatory and major histocompatibility complex molecules via Toll-like receptor 4, myeloid differentiation primary response protein 88, and Toll/IL-1 receptor-domain containing adaptor inducing IFN-beta [125]. Mitogen-activated protein kinases and NF-κB signaling pathways were involved in Rv2882c-induced macrophage activation. Further, we showed that Rv2882c-treated macrophages induce expansion of the effector/memory T cell population and Th1 immune responses. In addition, boosting Bacillus Calmette-Guerin vaccination with Rv2882c improved protective efficacy against *Mtb* in our model system [125]. Similar to Rv2882c, we identified another *Mtb* protein, Rv2005c, which induced DC maturation and Th1 polarisation, thus eliciting antimycobacterial activity in macrophages under hypoxic conditions [126]. Interestingly, although Rv2005c did not show any significant vaccine efficacy in a mouse model, its fusion to the macrophage activating Rv2882c produced a conjugated antigen with a strong therapeutic potential as adjunctive chemotherapy in a mouse model of TB infection [126]. Also, the expression of Rv2005c, which is a DosR regulon-encoded protein, is increased during the dormancy and reactivation of *Mtb* [144].

Rv2882c plays a role as a ribosome recycling factor (RRF), helping ribosomes to dissociate from mRNA in the post-termination complex, thus allowing ribosome recycling [145]. Typical of RRFs, Rv2882c has an overall L-shaped structure made of two domains with a high degree of flexibility [145]. No structural information is instead available for Rv2005c (Table 2) [145]. Using AI, we predicted the structure of the conjugated antigen with high reliability (pLDDT > 90). As shown in Figure 7, the Rv2005c portion of the Rv2882c-Rv2005c fused antigen displays the typical structural features of a widely distributed family of the Universal Stress Protein (USP) superfamily [146]. Indeed, using a structural alignment with the software DALI, we found more than 100 USP protein structures in the PDB with high structural similarity to Rv2005c (DALI Z score > 10). This structural information will be important for future studies aimed at improving the antigen immunomodulating properties.

### 5.3. Rv2299c and ESAT6: Combining Dendritic and T Cell Stimulation

Rv2299c, also denoted also as HtpG_Mtb_, belongs to the heat-shock protein 90 (HSP90) family and was identified as a potent DC-maturation agent [130,131]. The protein is highly conserved among *Mtb* strains [129], a finding that makes HtpG_Mtb_ an exceptional candidate for vaccines which target Th1 immunity. This type of response is essential in TB infection and highly depends on the early activation of DCs and their migration to the lymph nodes, where they can efficiently present antigens and stimulate proliferation of T cells [131]. HtpG_Mtb_ activates DCs due to TLR4 activation, with DC maturation accompanied by an enhancement of MHCI and MHCII expression, and elevated production of IL-6, IL-1β, TNF-α and IL-12p70, but not IL-10. DCs treated in vitro with HtpG_Mtb_ are able to stimulate proliferation of naïve-T cells toward Th1 phenotype, increasing their IFN-γ and IL-2 production [131]. Moreover, proliferation induced by HtpG_Mtb_-matured DCs significantly increased Th1 inhibition of intracellular *Mtb* growth in infected. Interestingly, better effects were obtained using HtpG_Mtb_ fused to ESAT-6, a well-known T cell activator. This conjugated antigen was also able of boosting effect in mice previously vaccinated with BCG [131].

Although no experimental structures of HtpG_Mtb_ are still available, computational approaches showed that HtpG_Mtb_ presents a highly stable dimeric structure, where each monomer is composed of three domains [130]. A large conformational flexibility was associated to the proposed clamping mechanism needed for the chaperon function of HtpG_Mtb_. Moving from an open state to a collapsed state, HtpG_Mtb_ can efficiently bind proteins to fold, and then release it when the folding process is finished [129,130,147] (Figure 8A,B). Using a SV approach, we predicted and experimentally confirmed that the most immunoreactive regions are located on the C-terminal and middle domains, whereas the N-terminal catalytic domain plays no role in elicitation of the immune response. These findings led us to design of new vaccine antigen candidates with enhanced biophysical properties and easier to produce, albeit conserved or enhanced antigenic properties [129]. Importantly, the antigen obtained upon conjugation of the C-terminal and Middle domains with ESAT6 (HtpG-MC_Mtb_-ESAT6), Figure 8C, was more immunoreactive than the full-length HtpG_Mtb_-ESAT6. Indeed, T cells activated by DCs matured with HtpG-MC_Mtb_-ESAT6 induced best *Mtb* growth inhibition in macrophages [129].

## 6. Conclusions and Perspectives

A huge scientific advance has been made since the discovery of the BCG vaccine against TB, in 1921. Meanwhile, and however, TB has undergone phases when there were reasons to believe that it was a thing of the past. Indeed, antibiotics developed since 1940s and 1950s led to a dramatic reduction in TB case fatality, to 5 percent or less, where used correctly. In 1970s, the introduction of rifampicin in the TB formulation reduced treatment to six/eight months, a treatment known as “short-course”. The TB burden seemed to be vanquished.

In 1993, WHO’s Global Tuberculosis Programme (GTB) declared TB a global emergency and launched the Directly observed treatment Strategy (DOTS), due to the development of antibiotic resistant mycobacterial strains. The concept of DOTS was to prevent the development of drug-resistant strains of TB, often fatal and almost 100 times more expensive to cure. However, 30 years later prevention is still scarcely approached, as BCG is still the only licensed vaccine against TB and its effectiveness is questionable.

A fair analysis of literature makes it evident that a strong acceleration has characterised the last two decades, as over sixteen vaccine candidates are currently in clinical phases. Of these, five are subunit vaccine antigens. Also, there are strong reasons to believe that this acceleration will further proceed, given the huge impact of bioinformatics and structural biology on vaccine development, e.g., to enhance the immune response to subunit antigens and rationally design conjugation strategies to contemporarily exploit multiple immune responses. We reckon that the combination of SV to identify immunogenic regions and conjugation strategies to induce immune modulating mechanisms are the keys to future acceleration in vaccine development. In this framework, we present in this review examples of vaccine antigens studied in our labs, which have benefited and will further benefit of SV for their improvement. To speed up this process is our increasing awareness of the structural basis of immunogenicity, an important information to design tailored antigens with improved biophysical properties and able to induce multiple immune stimulating mechanisms.

## Figures and Tables

**Figure 1 cells-12-00317-f001:**
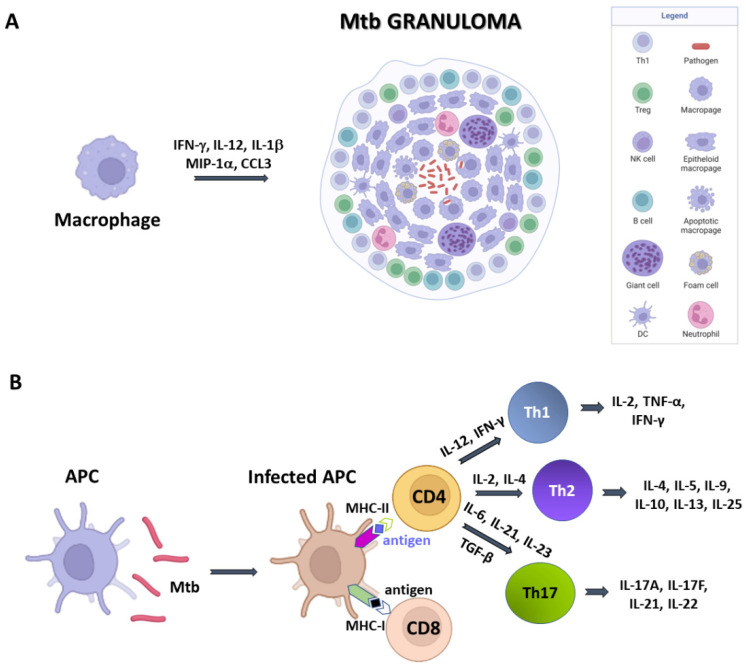
A simplified sketch of *Mtb*-induced immune reaction. (**A**) macrophages produce cytokines which recruit other immune cells to form the *Mtb* granuloma. In these structures, macrophages can differentiate in other cell types, like epithelioid cells, giant cells, and foamy macrophages. (**B**) Antigen presenting cells (APCs) expose antigens on their MHC molecules, thus bringing them to T cells CD4 and CD8. Cytokine production by APCs induce T cell differentiation. Created with BioRender.com, accessed on 20 December 2022.

**Figure 2 cells-12-00317-f002:**
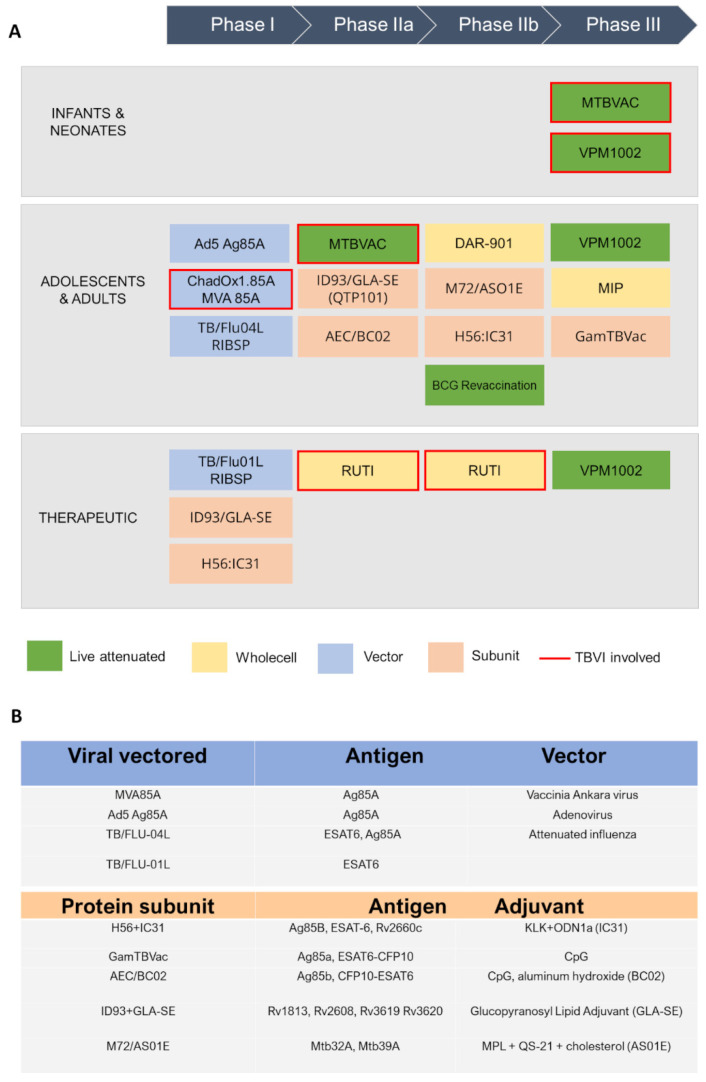
A schematic view of TB antigens in clinical trials. (**A**) Summary of TB vaccine candidates currently in clinical trials, updated to October 2022 (https://www.tbvi.eu/what-we-do/pipeline-of-vaccines/, accessed on 20 October 2022). (**B**) specific antigens encoded in viral vectored vaccines (top) and contained in protein subunit vaccines, along with their adjuvants (bottom).

**Figure 3 cells-12-00317-f003:**
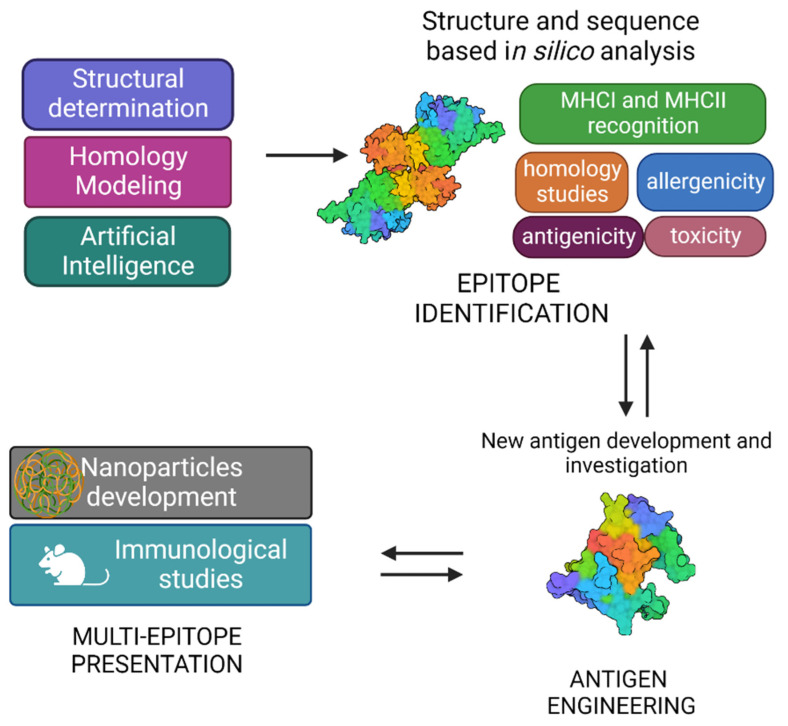
Graphical representation of structural vaccinology approach.

**Figure 4 cells-12-00317-f004:**
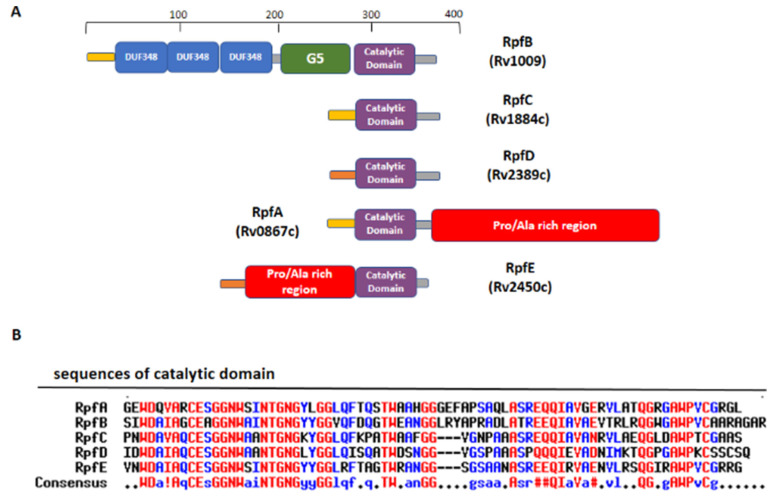
Domain composition of *Mtb* Rpfs. (**A**) A schematic view of domain composition in RpfA-E. (**B**) Sequence alignment of catalytic domains of the five Rpfs, computed with the MultAlin tool (high- and the low-consensus residues are indicated in red and blue, respectively).

**Figure 5 cells-12-00317-f005:**
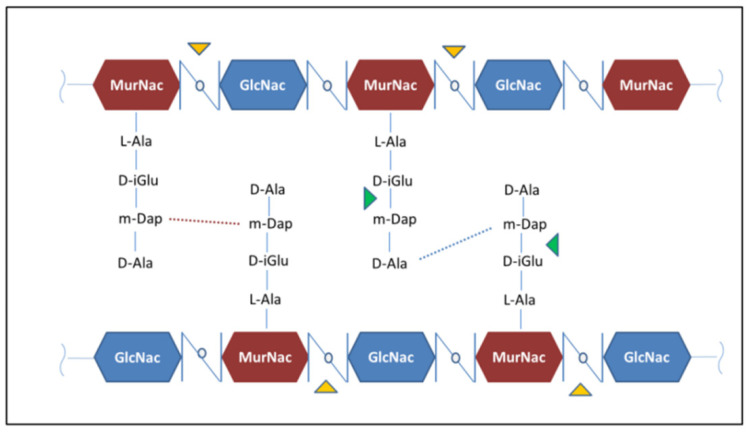
The chemical structure of *Mtb* peptidoglycan. Hydrolysis sites operated by Rpfs and RipA are highlighted by the yellow and green triangles, respectively.

**Figure 6 cells-12-00317-f006:**
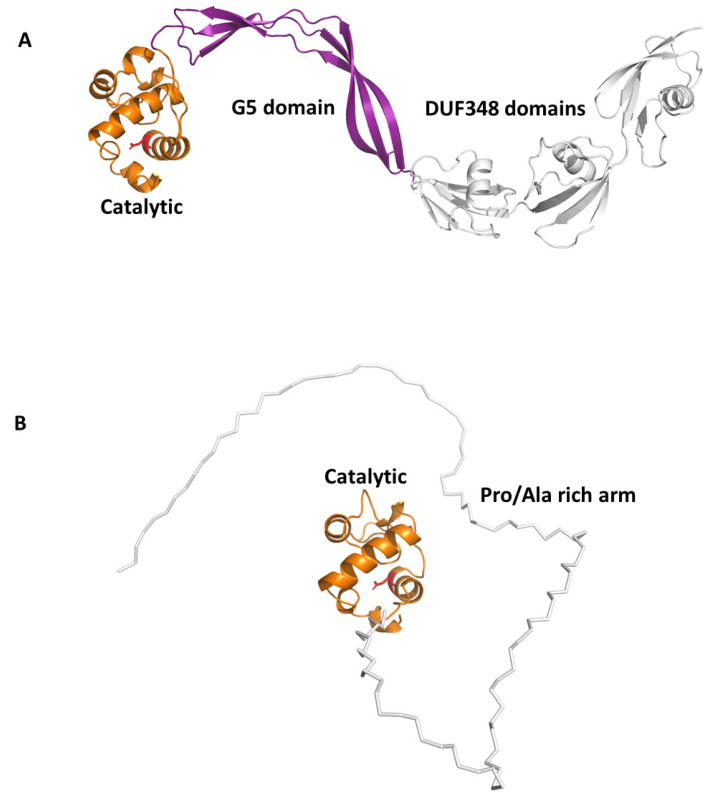
AlphaFold structures of antigenic Rpfs. (**A**) Cartoon representation of RpfB computed in a template-based mode using the crystal structure 5e27 as a template [122,123]; catalytic, G5 and DUF348 domains are drawn in orange, purple and grey, respectively. (**B**) Cartoon representation of the reliable region of RpfE structure, the catalytic domain in orange. The Pro/Ala rich arm, depicted using a ribbon representation, is highly flexible and was computed by AlphaFold with low reliability (pLDDT < 70).

**Figure 7 cells-12-00317-f007:**
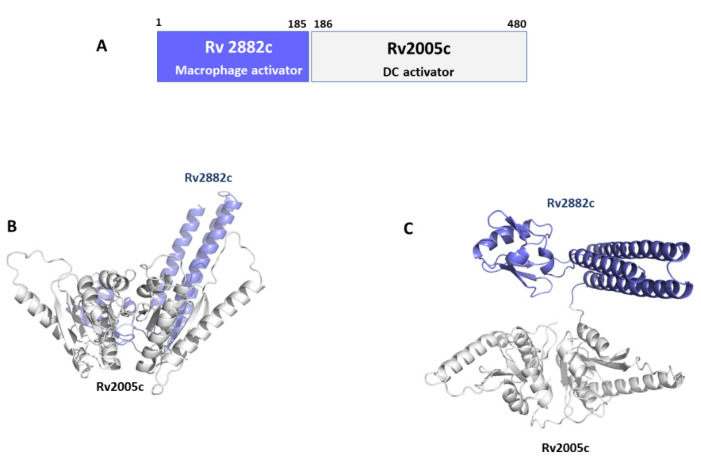
Conjugated antigen Rv2882c-Rv20025c. (**A**) Conjugation strategy, combining the macrophage activator Rv2882c with the DC activator Rv2005c. Cartoon representations of AlphaFold modelled structure of the conjugated antigen are shown in (**B**) side- and (**C**) top-views.

**Figure 8 cells-12-00317-f008:**
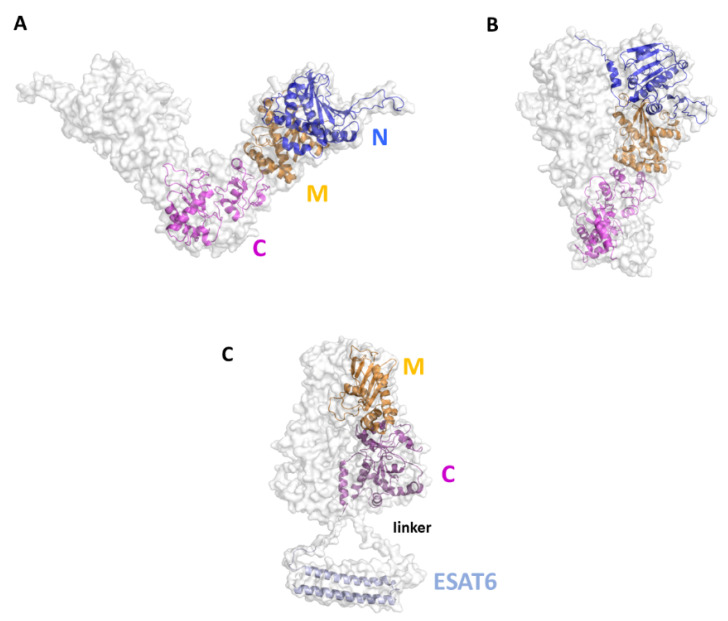
Cartoon and surface representations of the HtpG_Mtb_ antigen. Homology models of the open (**A**) and close (**B**) conformations of HtpG_Mtb_. Each chain is composed of the three domains N (residues 12–246), M (247–416) and C (417–645). (**C**) Model of the conjugated HtpG-MC_Mtb_-ESAT6. A linker was inserted between the C domain and ESAT6 (sequence PNSSSVDKL, [129]). The conjugation strategy combines the DC activating property of HtpG-MC_Mtb_ with the T-cell activating property of ESAT6.

**Table 2 cells-12-00317-t002:** A selected panel of subunit antigens against *Mtb*.

*Mtb* Protein	Putative Function	Immunogenic Function	Localisation	Structural Information (pdb Code)	Reference
Rv2450c (RpfE)	Cell Wall Hydrolase	-DC maturation	periplasm	4CGE	[119,120]
Rv1009 (RpfB)	Cell Wall Hydrolase	-DC maturation	periplasm	4KL7, 4KPM, 4EMN, 3EO5, 5E27	[121,122,123,124]
Rv2882c	Ribosome recycling factor	-Macrophage- activator -Boosting BCG	cytoplasm	4KAW, 4KB2, 4KB4, 4KC6, 4KDD	[125]
Rv2005c	USP	-DC maturation when fused to Rv2882c	cytoplasm	No structure available	[126]
Rv3463	Oxidoreductase	-Macrophage- activator	cytoplasm	No structure available	[127]
Rv1876	Bacterioferritin	-DC maturation -Boosting BCG	cytoplasm	3UOF, 3QB9, 3UOI	[128]
Rv2299c	Mycobacterial Chaperone	-DC maturation -Boosting BCG fused with ESAT6	cytoplasm	No structure available	[129,130,131]

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
