# Peer review of "A Structural View at Vaccine Development against M. tuberculosis"

_cells, 2023, doi:10.3390/cells12020317_

Round 1

Reviewer 1 Report

The authors present a fantastic perspective of the current knowledge on the required structural features to develop vaccines against M. tuberculosis. The chosen references are excellent, the style is concise, the discussion is easy to read also for non specialists, and the figures and Tables very well designed. Overall, excellent and profound.

Author Response

We thank the referee for her/his extremely supportive comment.

Reviewer 2 Report

​​This review provides an interesting overview of the current stage of TB vaccine development and the potential role of Structural Vaccinology in further improving subunit vaccines. The authors have done a great job in providing a comprehensive overview of the various clinical phases and strategies for vaccine development, as well as the contribution of Structural Vaccinology to the development of safer and effective antigens. The authors have also provided a detailed description of the host immune response to TB infection, which is key to the design of effective antigens. However, there are some areas that could be further explored. 

1. It would be interesting to discuss how Structural Vaccinology can be used to develop vaccines against drug-resistant strains of TB with high specificity without disturbing microbiome.

2. This review provides an overview of the current stage of TB vaccine development, but could benefit from a more comprehensive discussion of the challenges and limitations of the various vaccine technologies. 

3. The review could be improved by providing more information on the use of bioinformatic tools and ML-based methods to predict B- and T-cell epitopes, evaluate their conservation among other pathogenic antigens, and predict immunogenicity, toxicity and solubility. 

4. The review could be strengthened by providing more detail on the use of experimental and computational epitope mapping to design multi-epitope vaccines against TB. Overall, this review provides an informative overview of TB vaccine development and the potential role of Structural Vaccinology in further improving subunit vaccines. With some additional information on some areas, this review could be even more comprehensive.

Author Response

General comment

This review provides an interesting overview of the current stage of TB vaccine development and the potential role of Structural Vaccinology in further improving subunit vaccines. The authors have done a great job in providing a comprehensive overview of the various clinical phases and strategies for vaccine development, as well as the contribution of Structural Vaccinology to the development of safer and effective antigens. The authors have also provided a detailed description of the host immune response to TB infection, which is key to the design of effective antigens. However, there are some areas that could be further explored. 

Comment

  1. It would be interesting to discuss how Structural Vaccinology can be used to develop vaccines against drug-resistant strains of TB with high specificity without disturbing microbiome.

Response

We thank the referee for her/his supportive comment and for giving us the chance to discuss this interesting point. The influence of microbiota on vaccination is a stimulating field of science. We have now added two important references (92 and 93). To answer the referee’s question, we have included a new paragraph at p. 10, which reads:

“While designing an antigen, a comprehensive sequence analysis is needed to evaluate the level of antigen conservation among serotypes and, in luckiest cases, among different pathogenic bacteria. During this process, an effort has to be made to skip those antigens which are also conserved in non-pathogenic strains, especially those belonging to the microbiota. Indeed, it has been recently shown that the composition and function of the gut microbiota are crucial in modulating immune responses to vaccination [92], especially in early life immunity [93].”

Comment

  1. This review provides an overview of the current stage of TB vaccine development but could benefit from a more comprehensive discussion of the challenges and limitations of the various vaccine technologies. 

Response

We thank the referee for this comment. We had already shortly discussed about challenges and limitations in the original version of the manuscript, but we agree that further discussion would be of benefit. Therefore, we have added extra sentences in chapter 3. For each type of vaccine (from 3.1 to 3.4), we included advantages and drawbacks (wordtrack).

Comment

  1. The review could be improved by providing more information on the use of bioinformatic tools and ML-based methods to predict B- and T-cell epitopes, evaluate their conservation among other pathogenic antigens, and predict immunogenicity, toxicity and solubility. 
  2. The review could be strengthened by providing more detail on the use of experimental and computational epitope mapping to design multi-epitope vaccines against TB. Overall, this review provides an informative overview of TB vaccine development and the potential role of Structural Vaccinology in further improving subunit vaccines. With some additional information on some areas, this review could be even more comprehensive.

Response

The referee is right, although it is not easy to be exhaustive in describing these wide methodologies. We have added an extra Table (Table 1, in the revised version) with most common bioinformatic tools. As for sequence conservation, we also replied in the comment #1. 

Reviewer 3 Report

The work entitled:" A structural view at vaccine development against M. tuberculosisdone by Romano etal is well presented and is of great importance however, there are some points should be addressed before being published.

1- in the introduction, the research question and the research hypothesis are need to be clarified.

2- In introduction, no enough information about previous works.

3- In introduction, the author need to clarify the challenges more.

4- In vaccine approaches, clarify the different candidates in details

5- Little English errors, please go over the manuscript.

Author Response

The work entitled:" A structural view at vaccine development against M. tuberculosis" done by Romano etal is well presented and is of great importance however, there are some points should be addressed before being published.

1- in the introduction, the research question and the research hypothesis are need to be clarified.

2- In introduction, no enough information about previous works.

3- In introduction, the author need to clarify the challenges more.

4- In vaccine approaches, clarify the different candidates in details

5- Little English errors, please go over the manuscript.

Response

We thank the referee for her/his supportive comments. We have modified the Introduction to address the referee’s concerns. We have also fixed a few English errors, as requested.